# Spontaneous spin-valley polarization in NbSe$_2$ at a van der Waals interface

Hideki Matsuoka[1,2], Tetsuro Habe[3,4], Yoshihiro Iwasa [1,2], Mikito Koshino[5] & Masaki Nakano [1,2] ✉

A proximity effect at a van der Waals (vdW) interface enables creation of an emergent quantum electronic ground state. Here we demonstrate that an originally superconducting two-dimensional (2D) NbSe$_2$ forms a ferromagnetic ground state with spontaneous spin polarization at a vdW interface with a 2D ferromagnet V$_5$Se$_8$. We investigated the anomalous Hall effect (AHE) of the NbSe$_2$/V$_5$Se$_8$ magnetic vdW heterostructures, and found that the sign of the AHE was reversed as the number of the V$_5$Se$_8$ layer was thinned down to the monolayer limit. Interestingly, the AHE signal of those samples was enhanced with the in-plane magnetic fields, suggesting an additional contribution to the AHE signal other than magnetization. This unusual behavior is well reproduced by band structure calculations, where the emergence of the Berry curvature along the spin-degenerate nodal lines in 2D NbSe$_2$ by the in-plane magnetization plays a key role, unveiling a unique interplay between magnetism and Zeeman-type spin-orbit interaction in a non-centrosymmetric 2D quantum material.

Van der Waals (vdW) superstructures constructed with a variety of two-dimensional (2D) materials play a central role in modern condensed matter physics and materials science[1,2], providing a perspective that has never been discussed so far. A remarkable example is an emergent strong correlation effect in a moiré flat band system made of weakly correlated materials such as graphene and semiconducting transition-metal dichalcogenides (TMDCs)[3–5], where the motion of electrons is quenched and the low-energy electronic properties are governed by the Coulombic interaction. Another intriguing example is a strong proximity effect at a vdW interface composed of atomically thin quantum materials, where the hybridization of electron wavefunctions leads to generation of a novel quantum electronic ground state that is missing in individual materials, as exemplified by the emergent topological states observed in the graphene/TMDC heterostructures as well as in the NbSe$_2$/CrBr$_3$ heterostructures[6–8]. Those studies also verify a strong electronic coupling at a vdW interface despite that the constituent materials are weakly bonded via the vdW

force, suggesting that a vdW interface should provide an ideal material platform for designing and creating novel physical properties and functionalities.

Among various possibilities, fabrication of a magnetic vdW heterostructure is an interesting and important research direction both for fundamental and applied researches[2,9,10]. Recent studies on the WSe$_2$/CrI$_3$ heterostructures have revealed that a magnetic exchange interaction is present at a vdW interface even though each layer is well separated by a vdW gap[9,10], providing the so-called "valley-Zeeman effect"[11–13] at zero field in WSe$_2$ induced by the exchange field from neighboring CrI$_3$ across the vdW interface. In this system, however, the ground state of WSe$_2$ should be still non-magnetic without spontaneous spin polarization, because WSe$_2$ is a semiconductor characterized with the fully occupied band, where the numbers of the up-spin and down-spin electrons below the Fermi level ($E_F$) should be always equal even when proximitized by a ferromagnet. On the other hand, a magnetic proximity effect in a metallic system characterized with the

[1]Quantum-Phase Electronics Center and Department of Applied Physics, the University of Tokyo, Tokyo 113-8656, Japan. [2]RIKEN Center for Emergent Matter Science (CEMS), Wako 351-0198, Japan. [3]Division of Applied Physics, Hokkaido University, Hokkaido 060-0808, Japan. [4]Nagamori Institute of Actuators, Kyoto University of Advanced Science, Kyoto 615-0096, Japan. [5]Department of Physics, Osaka University, Osaka 560-0043, Japan. ✉e-mail: nakano@ap.t.u-tokyo.ac.jp

partially occupied band should lead to generation of a novel magnetic ground state with spontaneous spin polarization by producing an imbalance between the numbers of the up-spin and down-spin electrons below $E_F$. Moreover, such a metallic system should enable us to examine the low-energy electronic properties of the system by the anomalous Hall effect (AHE) measurements, through which we should be able to obtain fundamental information on the electronic structure of the system near $E_F$. Taken together, studying a magnetic proximity effect in a metallic vdW system should be of significant importance, but such an attempt has not been reported so far.

In this study, we demonstrate a proximity-induced ferromagnetic ground state in atomically thin $NbSe_2$ at a vdW interface with a 2D ferromagnet, and uncover an intriguing feature of ferromagnetic $NbSe_2$ associated with its unique band structure by the AHE measurements. $NbSe_2$ is one of representative metallic $H$-type TMDCs showing the charge-density wave (CDW) and the superconducting (SC) transition at low temperature[14], while it is magnetically inactive due to highly delocalized nature of $4d$ electrons in $NbSe_2$. An unprecedented feature of $NbSe_2$ in the context of 2D materials research arises when thinned down to the monolayer limit, where broken in-plane inversion symmetry relevant to the trigonal prismatic structure (see Fig. 1a) and large spin-orbit interaction (SOI) lift the spin degeneracy near $E_F$. This results in generation of the out-of-plane spin-polarized electrons near $E_F$ as schematically illustrated in Fig. 1b[15–17], while the system is non-magnetic due to time-reversal symmetry. Such SOI is termed as "Zeeman-type SOI" or "Ising-type SOI", which plays an essential role in the spin-valley locking effect in monolayer $H$-type TMDCs, providing intriguing physical phenomena both in semiconducting and metallic $H$-type TMDCs[18,19]. On the other hand, an interplay between Zeeman-type SOI and magnetism has not been unveiled yet.

As for a 2D ferromagnet, we employed $V_5Se_8$, a layered magnet characterized by the 2D $VSe_2$ sheets separated by the layers of the periodically aligned V atoms (see Fig. 1c). This compound is known to be an itinerant antiferromagnet in the bulk form[20], whereas it exhibits weak itinerant ferromagnetism with negative AHE in the thin film form, which survives down to the 2D limit[21]. We recently succeeded in constructing a magnetic vdW heterostructure with an atomically abrupt vdW interface based on $NbSe_2$ and $V_5Se_8$ by molecular-beam epitaxy (MBE), and demonstrated that there is a strong proximity effect working at this $NbSe_2/V_5Se_8$ (Nb/V) interface characterized by the robust out-of-plane magnetic anisotropy and the large enhancement of the transition temperature[22]. Those results strongly suggest that there should be an interface exchange interaction present in the Nb/V heterostructures and that the electronic state of $NbSe_2$ should be also modulated, although no clear evidence is provided. Here, we systematically investigated the magneto-transport properties of the Nb/V heterostructures, where the number of the $V_5Se_8$ layer ($N$, varied) was set to be thinner than that of the $NbSe_2$ layer (4 L, fixed) so that the transport properties are dominated by $NbSe_2$. The details of the sample fabrication and characterization are described in our previous study[22], and brief summary is shown in Methods and Supplementary Note 1.

## Results

### The AHE of the NbSe₂/V₅Se₈ van der Waals heterostructures

Figure 2a shows the AHE data taken at $T = 2$ K for the series of samples, where $N$ was systematically reduced from 6 L to 1.2 L. The $N = 6$ L sample exhibited negative AHE with clear magnetic hysteresis loop, indicating the robust out-of-plane magnetic anisotropy induced in $V_5Se_8$ by a strong proximity effect from neighboring $NbSe_2$ as we discussed in the previous study[22]. On the other hand, as $N$ was decreased, the AHE signal was at first suppressed at around $N = 3.0$ L and then developed again with the opposite sign below $N < 3$ L. Such a sign change has never been observed in the $V_5Se_8$ individual films down to the 2D limit[21], implying essential contribution from $NbSe_2$.

The evolution of the AHE signals depending on $N$ was more systematically investigated by the temperature-dependence measurements. Figure 2b shows the temperature dependence of the AH resistance at the saturated regime ($R_{AH, sat}$) for the different $N$ samples. For the thick-enough regime ($N = 6$ L, for example), negative AHE developed monotonously below the transition temperature (~30 K). This behavior is basically the same as those of the $V_5Se_8$ individual films (although the transition temperatures are largely enhanced for the heterostructures as mentioned above)[21,22], suggesting that negative AHE observed with thick-enough $V_5Se_8$ should be attributed to originally ferromagnetic $V_5Se_8$. On the other hand, as $N$ was decreased, a positive component started to develop below 25 K, and the sign of the AHE at the lowest temperature was reversed for $N < 3$ L. Considering that the 3 L-thick $V_5Se_8$ individual film shows rather insulating behavior at low temperature (in particular below 10 K) while the 4 L-thick $NbSe_2$ individual film exhibits metallic behavior even below 2 K[21,23], it is natural to consider that the electrical conductions of the $N < 3$ L samples at the low temperature regime are predominantly governed by $NbSe_2$ (see Supplementary Note 2). Given that the AHE is absent in a non-magnetic material, the obtained results suggest that $NbSe_2$ is in a ferromagnetic state, and positive AHE observed with thin-enough $V_5Se_8$ should be attributed to ferromagnetically proximitized $NbSe_2$. The sign reversal of the AHE signal shown in Fig. 2a should be therefore originating from the fact that the dominant layer providing larger contribution to the electrical conduction is varied from $V_5Se_8$ to $NbSe_2$ as the number of the $V_5Se_8$ layer is reduced.

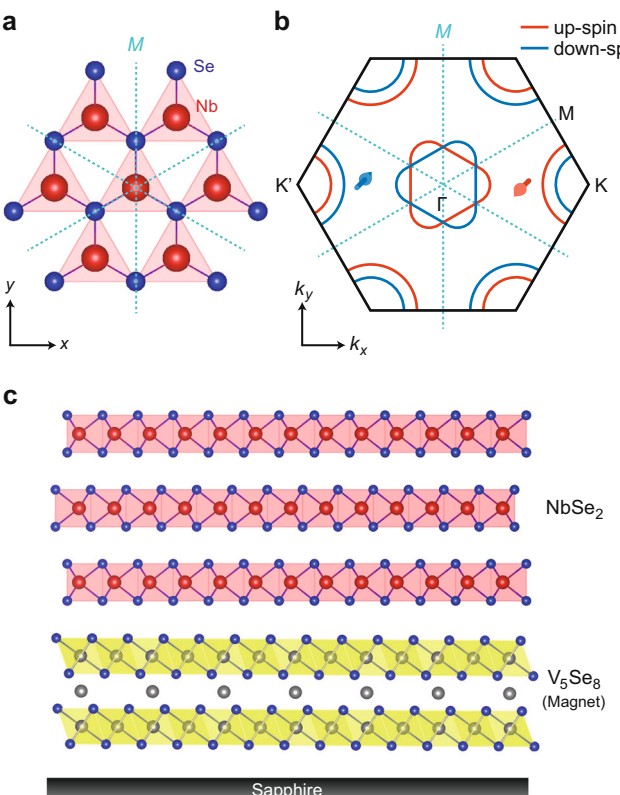

**Fig. 1 | A magnetic van der Waals interface made of NbSe₂ and V₅Se₈.**
**a** Schematic top view of $NbSe_2$ crystal. The dashed blue lines labeled with $M$ represent the mirror planes. **b** Schematic Fermi surface (FS) of monolayer $NbSe_2$ with Zeeman-type spin-orbit interaction (SOI). The red and blue solid lines denote the FS of the up-spin (red arrow) band and the down-spin (blue arrow) band, respectively. **c** Schematic side view of the $NbSe_2/V_5Se_8$ (Nb/V) heterostructure.

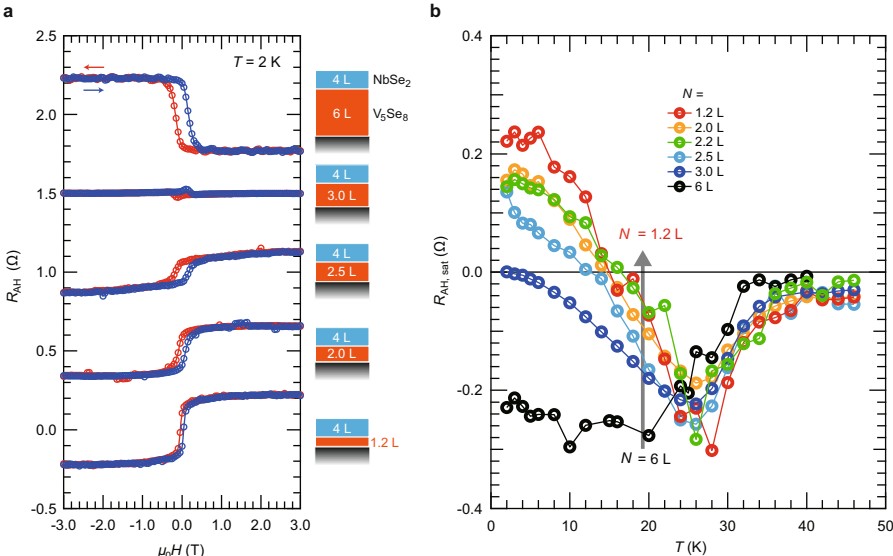

**Fig. 2 | The anomalous Hall effect (AHE) of the NbSe$_2$/V$_5$Se$_8$ van der Waals heterostructures. a** The anomalous Hall resistance ($R_{AH}$) of the Nb/V heterostructures as a function of the magnetic field ($\mu_0 H$) taken at $T = 2$ K, where the number of the V$_5$Se$_8$ layer ($N$) was varied while that of the NbSe$_2$ layer was fixed to 4 L. The $N$ is defined as the number of the host VSe$_2$ layer. The data of the $N = 6$ L, 3.0 L, 2.5 L, and 2.0 L samples are vertically shifted by 2.0 Ω, 1.5 Ω, 1.0 Ω, and 0.5 Ω, respectively. **b** The temperature dependences of the anomalous Hall resistance at the saturated regime ($R_{AH, sat}$) for the different $N$ samples.

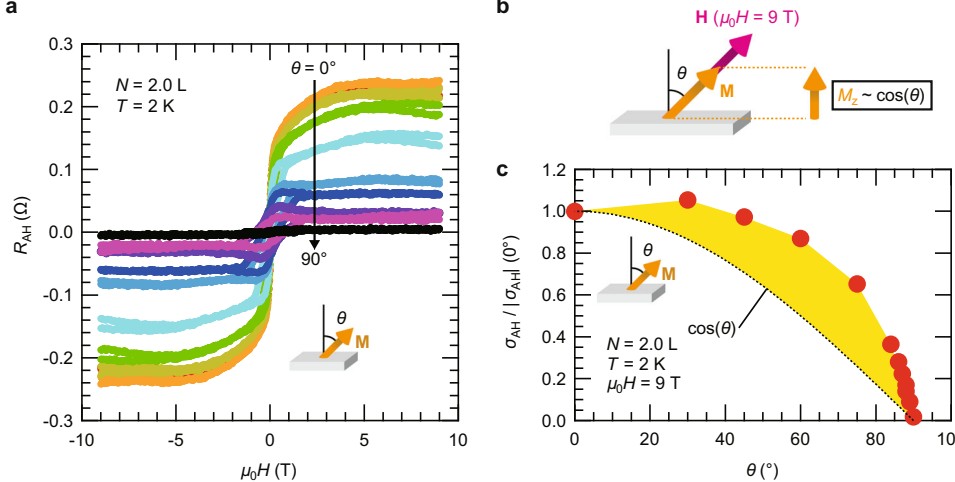

**Fig. 3 | The angle dependence of the AHE of the NbSe$_2$/V$_5$Se$_8$ heterostructure. a** The AHE of the $N = 2.0$ L sample at $T = 2$ K with different field angles ($\theta$). The configuration of the field angle $\theta$ is shown in the inset. **b** A schematic of the magnetization direction (**M**) and the field direction (**H**) at the high-enough field regime ($\mu_0 H = 9$ T). **c** The AH conductivity ($\sigma_{AH}$) at $\mu_0 H = 9$ T plotted against $\theta$. The data is normalized by the $\sigma_{AH}$ at $\theta = 0°$. The black dotted line represents cos ($\theta$) relative to the $\sigma_{AH}$ at $\theta = 0°$, and the yellow-colored area highlights a deviation from cos ($\theta$).

## The angle dependence of the AHE of the NbSe$_2$/V$_5$Se$_8$ heterostructure

The AHE arising from NbSe$_2$ is quite surprising in view of the fact that NbSe$_2$ is a well-known superconductor associated with CDW, where no experimental signatures of magnetism have been discussed so far. To get insights into the origin of this novel AHE in the NbSe$_2$/V$_5$Se$_8$ heterostructure system, we performed further experiments on the angle dependence of the AHE for those samples exhibiting positive AHE. Figure 3a displays the AHE of the $N = 2.0$ L sample taken at $T = 2$ K with different field angles ($\theta$). We here focus on the high-enough field regime, where the magnetization direction is fixed to the applied field direction. Given that the AHE signal is proportional to the out-of-plane component of the total magnetization, the signal should be reduced when the field is tilted to the in-plane direction by following cos ($\theta$) as schematically illustrated in Fig. 3b. However, the AHE in the present

system did not follow this expected behavior. Figure 3c shows the normalized AHE signal as a function of $\theta$ taken at $\mu_0 H = 9$ T, demonstrating a substantial deviation from cos ($\theta$) as highlighted by the yellow-colored area. This suggests that there is an additional contribution to the AHE signal other than magnetization arising when the in-plane magnetization becomes finite, which has never been observed in other 2D quantum material systems. Remarkably, as will be discussed in the following sections, this intriguing angle dependence as well as the positive sign of the AHE signal could be well reproduced by theoretical calculations based on the band structure of ferromagnetically proximitized NbSe$_2$, suggesting their intrinsic origins rather than the extrinsic origins associated with the scattering mechanisms. Moreover, such a good agreement between experiments and theory strongly suggests that NbSe$_2$ forms a ferromagnetic ground state with spontaneous spin polarization at the NbSe$_2$/V$_5$Se$_8$ vdW interface.

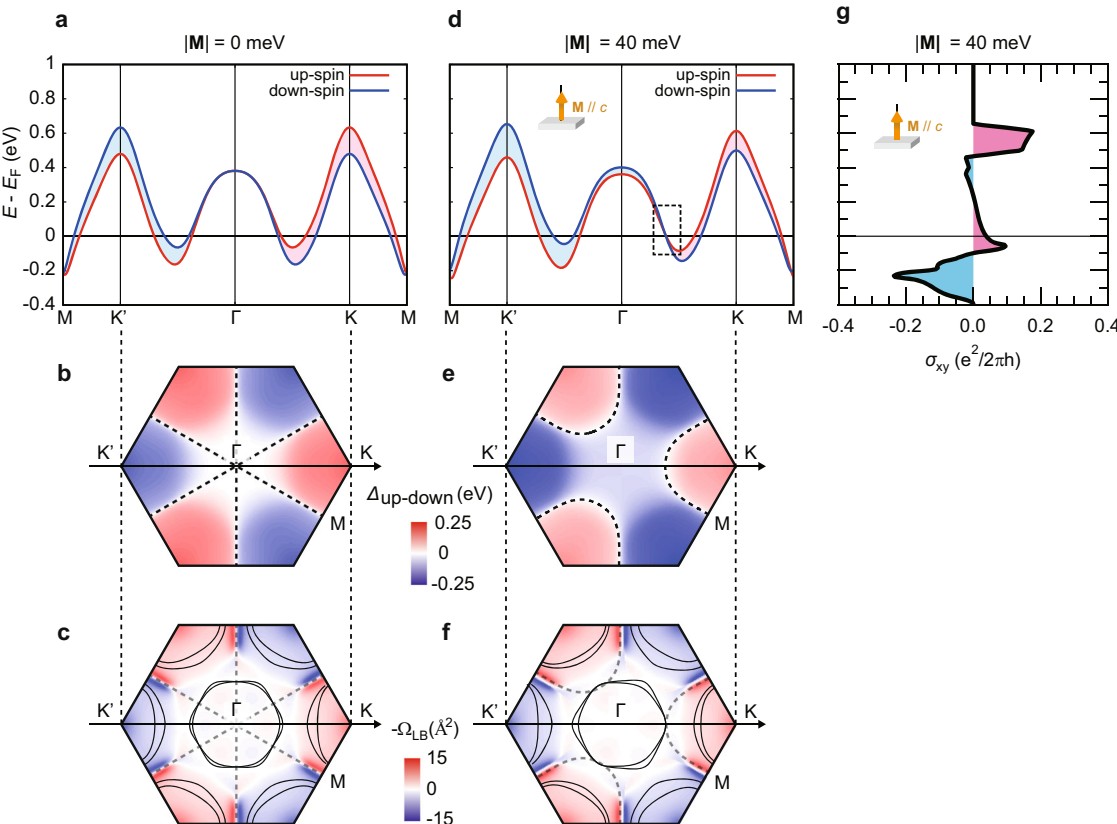

**Fig. 4 | Calculations of the band structure of ferromagnetic NbSe₂. a**, **d** The band structure of monolayer NbSe₂ (**a**) without and **d** with the exchange field (|**M**| = 40 meV) applied parallel to the *c*-axis (**M**//*c*). The dotted rectangular region in **d** corresponds to the area highlighted in Fig. 5b, d. **b**, **e** The magnitude of the spin splitting energy induced by Zeeman-type SOI ($\Delta_{\text{up-down}} = E_{\text{up}} - E_{\text{down}}$) in the first Brillouin Zone (BZ) (**b**) without and **e** with the exchange field. The dashed lines correspond to the spin-degenerate nodal lines. **c**, **f** The FS (black solid line) and the Berry curvature of the lower band ($\Omega_{\text{LB}}$ (**k**)) in the first BZ (**c**) without and **f** with the exchange field. The dashed lines represent the nodal lines. **g** The $\sigma_{xy}$ as a function of energy calculated from the band structure of monolayer NbSe₂ with $M_z$ = 40 meV shown in **d**.

## Calculations of the band structure of ferromagnetic NbSe₂

Now we discuss the origin of the AHE arising from ferromagnetic NbSe₂. We here consider the band structure of monolayer NbSe₂ assuming that a magnetic proximity effect is limited to the single layer in contact with V₅Se₈, but a more realistic case with multilayer NbSe₂ is discussed in detail in Supplementary Note 7, which provides qualitatively the same results. Figure 4a depicts the band dispersion of monolayer NbSe₂ near $E_F$ along the Γ-K-M direction in the momentum space (and its time-reversal pair), showing spin splitting due to Zeeman-type SOI. Figure 4b maps the magnitude of this spin splitting energy ($\Delta_{\text{up-down}}$) in the first Brillouin Zone (BZ), which has three-fold symmetry as expected for the trigonal crystal structure. The magnitude of $\Delta_{\text{up-down}}$ at the K and K′ valleys is exactly the same but opposite in sign because of time-reversal symmetry. Figure 4c illustrates the Fermi surface (FS) of monolayer NbSe₂ and the Berry curvature of the lower band (i.e., the band having lower energy at finite **k**) in the first BZ, $\Omega_{\text{LB}}$ (**k**), both of which are symmetric against time-reversal operation. In such a case, the AH conductivity ($\sigma_{xy}$) calculated by the integration of the Berry curvature below $E_F$ over the entire BZ becomes exactly zero, and no AHE expected. Another important aspect is that the spin degeneracy is protected along the Γ-M direction due to mirror symmetry (see Fig. 1a) as shown by the dashed lines in Fig. 4b, c, which appears only at the Γ and M points in the line cut along the Γ-K-M direction (Fig. 4a).

Let us now discuss the band structure of ferromagnetic NbSe₂ by considering the situation when NbSe₂ is subjected to the out-of-plane exchange field. Figure 4d shows the band structure of monolayer NbSe₂ with the exchange field (|**M**| = 40 meV) applied parallel to the *c*-axis (**M**//*c*). Now, the up-spin band and the down-spin band are shifted to the opposite directions due to Zeeman effect, and the magnitude of $\Delta_{\text{up-down}}$ at the K and K′ valleys becomes different (see Fig. 4d). Accordingly, the FS becomes largely distorted (see Fig. 4f), and the numbers of the up-spin and down-spin electrons below $E_F$ become unequal, providing a ferromagnetic ground state with spontaneous spin polarization. In such a ferromagnetic state, we expect non-zero $\sigma_{xy}$ due to imperfect cancellation of the Berry curvature below $E_F$ between two bands.

Figure 4g shows the resultant $\sigma_{xy}$ as a function of energy calculated from the band structure of ferromagnetic NbSe₂, showing non-zero $\sigma_{xy}$ at $E = E_F$. Moreover, the sign of $\sigma_{xy}$ at $E = E_F$ is positive, which is consistent with the experimentally observed positive AHE for the $N < 3$ L samples, suggesting that positive AHE is indeed originating from ferromagnetic NbSe₂. Considering that the K and K′ valleys are now energetically inequivalent, this ferromagnetic state would possibly be interpreted as a "ferrovalley" state with spontaneous valley polarization as well[24], which is a natural consequence of the spin-valley locking effect in *H*-type TMDCs associated with Zeeman-type SOI[15–17]. We note that the spin-degenerate nodal lines shown as the dashed lines in Fig. 4b, c are now moved away from the Γ-M direction, forming the closed triangular loops surrounding the K valleys (see Fig. 4e, f). The corners of those loops are contacted with the FS near the Γ valley (see Fig. 4f), which is essentially important to explain the results of the angle-dependence measurements as will be discussed in the next section.

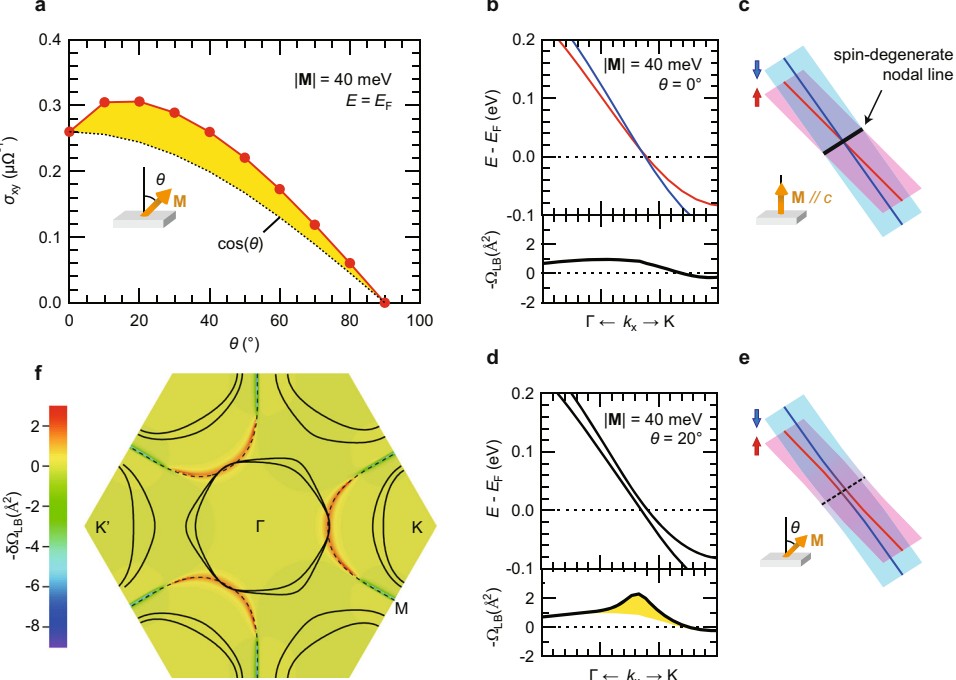

**Fig. 5 | Calculations of the angle dependence of the AHE of ferromagnetic NbSe₂.** **a** The calculated angle dependence of the $\sigma_{xy}$ with |**M**| = 40 meV at $E = E_F$. The black dotted line represents cos ($\theta$) relative to the $\sigma_{xy}$ at $\theta = 0°$, and the yellow-colored area highlights a deviation from cos ($\theta$). **b**, **d** The magnified views of the band structures and the Berry curvatures in the dotted rectangular region in Fig. 4d for **b** $\theta = 0°$ and (**d**) $\theta = 20°$. The absolute value of the exchange field is fixed to |**M**| = 40 meV. The yellow-colored area highlights the emergent Berry curvature generated by the in-plane magnetization. **c, e** The corresponding schematics of the band structures around the spin-degenerate nodal lines. **f** The distribution of the emergent Berry curvature at $\theta = 20°$ defined as $\delta\Omega_{LB}$ (**k**) = $\Omega_{LB}$ (**k**, 20°) − $\Omega_{LB}$ (**k**, 0°) in the momentum space. The dashed lines correspond to the nodal lines and the black solid lines illustrate the FS.

## Calculations of the angle dependence of the AHE of ferromagnetic NbSe₂

Figure 5a presents the angle dependence of $\sigma_{xy}$ at $E = E_F$ calculated from the band structure of ferromagnetic NbSe₂. Surprisingly, a deviation from cos ($\theta$) that we experimentally observed for the $N < 3$ L samples (see Fig. 3c) was successfully reproduced. This characteristic deviation from cos ($\theta$) could be considered as the enhancement of the AHE signal with the in-plane fields, which could be understood as a consequence of the emergence of the additional Berry curvature with the in-plane magnetization[25], where the intersection of the up-spin band and the down-spin band (i.e., the spin-degenerate nodal lines) associated with Zeeman-type SOI plays an essential role. As we explained above, those lines are originally located along the Γ-M direction without the exchange field (see Fig. 4b, c), while they are moved to surround the K valleys under the presence of the out-of-plane exchange field (see Fig. 4e, f). Importantly, the location of those nodal lines in the momentum space is determined by the magnitude of the out-of-plane magnetization ($M_z$), and $M_z = 40$ meV satisfies the condition that the nodal lines come close to the corner of the FS near the Γ valley (see Fig. 4d, f), providing the largest deviation from cos ($\theta$) as will be discussed later.

The top panel of Fig. 5b shows the magnified view of the band structure near one of such nodal lines, corresponding to the dotted rectangular region in Fig. 4d. There is only one crossing point visible in this plot, but this is in reality distributed in the momentum space to surround the K valleys as mentioned above (also schematically shown in Fig. 5c). When the magnetization direction is tilted to the in-plane direction (for example, $\theta = 20°$ is considered in Fig. 5d), the in-plane component of the magnetization ($M_{xy}$) becomes finite, which hybridizes the up-spin band and the down-spin band and opens an energy gap along the nodal lines as shown in Fig. 5d, e. Importantly, this gap opening accompanies the generation of the Berry curvature along the nodal lines as shown in the bottom panel of Fig. 5d by the yellow-

colored area. Figure 5f illustrates the distribution of this emergent Berry curvature at $\theta = 20°$ defined as $\delta\Omega_{LB}$ (**k**) = $\Omega_{LB}$ (**k**, 20°) − $\Omega_{LB}$ (**k**, 0°) in the momentum space, which exactly traces the nodal lines (loops) surrounding the K valleys. Also shown is the FS of monolayer NbSe₂ in a ferromagnetic state with the exchange field $M_z = 40$ meV, which is contacted with the peak of the emergent Berry curvature $\delta\Omega_{LB}$ (**k**) near the Γ valley. In this situation, the imbalance between $\sigma_{xy}$ arising from the lower band and that from the upper band becomes maximum, providing the largest deviation from cos ($\theta$). We however note that such a deviation from cos ($\theta$) is observable in a rather broad range of the exchange field as long as $E$ is fixed at $E_F$, whereas it disappears when $E$ is increased near to the valence band top (see Supplementary Note 6). This suggests that the observed phenomena associated with the emergence of the Berry curvature with the in-plane magnetization are unique to group-V metallic $H$-type TMDCs, and that the crossing of the FS and the nodal lines near the Γ valley plays a key role for the enhancement of the AHE signal with the in-plane fields.

## Discussion

The present study demonstrates a proximity-induced ferromagnetic ground state with spontaneous spin polarization in a metallic 2D quantum material NbSe₂ originally showing the CDW/SC states. We emphasize that the unique angle dependence of the AHE characterized with the enhancement of the AHE signal with the in-plane fields observed by experiments could be well reproduced by theoretical calculations based on the band structure of ferromagnetic NbSe₂, providing firm evidence that NbSe₂ forms a ferromagnetic ground state at the interface with V₅Se₈. Based on the comparison between the experimental and theoretical results, we estimated that the magnetic exchange interaction at the Nb/V interface should be as large as a few tens of millielectronvolt, which is larger than those reported for the WSe₂-based magnetic heterostructures[9,10,26] and comparable to the value recently reported for the WS₂-based heterostructures[27]. As for a

possible origin of the anomalous enhancement of the AHE signal with the in-plane fields, we interpret from theoretical consideration that the generation of the Berry curvature along the spin-degenerate nodal lines in 2D NbSe$_2$ by the in-plane magnetization should play an essential role, resulting from a unique interplay between magnetism and Zeeman-type SOI in 2D NbSe$_2$. Furthermore, recent demonstration of topological superconductivity in the NbSe$_2$/CrBr$_3$ heterostructures implies the possible existence of a topological superconducting state in our all-epitaxial scalable Nb/V heterostructures as well[8], which should provide an ideal platform for future topological quantum computing device applications.

## Methods

### Sample fabrication and characterization

All the Nb/V heterostructures were grown on commercially available insulating Al$_2$O$_3$ (sapphire) (001) substrates (SHINKOSHA Co., Ltd.) by MBE by following our previously established process[21–23]. A substrate was cleaned by ultrasonication in acetone and ethanol, respectively, annealed in air at 1000 °C for three hours to make a surface atomically flat, and then transferred into the ultrahigh vacuum chamber with a base pressure below ~1 × 10$^{-7}$ Pa (EIKO Engineering, Ltd.). Prior to the film growth, a substrate surface was treated with the Se flux at 900 °C for an hour. Then, the substrate temperature was decreased down to 450 °C, and the V$_5$Se$_8$ layer was grown at 450 °C. During the growth, V was supplied by an electron beam evaporator with the evaporation rate of ~0.01 Å/s, while Se was supplied by a standard Knudsen cell throughout the growth process with the rate of ~2.0 Å/s. After the growth of the V$_5$Se$_8$ layer, the NbSe$_2$ layer was formed at the same growth temperature. Nb was supplied by an electron beam evaporator with the rate of ~0.01 Å/s as well. The whole growth process was monitored by reflection high energy electron diffraction (RHEED). Clear RHEED intensity oscillations were observed during the growth, confirming the 2D layer-by-layer growth mode for the formation of the V$_5$Se$_8$ layer and the NbSe$_2$ layer. The exact thickness was designed by counting the number of the RHEED intensity oscillation, and confirmed by the x-ray diffraction (XRD) measurement (PANalytical, X'Pert MRD) after the growth. The local structure of the obtained heterostructure was characterized by scanning transmission electron microscopy (STEM) measurement (JEOL, JEM-ARM200F). All the samples were covered by insulating Se capping layers to protect their surfaces from oxidization. See Supplementary Note 1 for the details.

### Transport measurements

All the samples were cut into Hall-bar shape before transport measurements by mechanical scratching through Se capping layers to define the channel regions, which were typically a few hundred micrometers. The electrical transport properties were characterized by Physical Property Measurement System (Quantum Design, PPMS).

### Theoretical calculations

The numerical calculations were performed by using density functional theory (DFT) and a multi-orbital tight-binding model. The electronic structure of monolayer NbSe$_2$ was obtained by using quantum-ESPRESSO[28], a package of numerical codes for DFT calculations, with the projector augmented wave method. The cut-off energy was set to 50 Ry for the plane wave basis and 500 Ry for the charge density. The convergence criterion of 10$^{-8}$ Ry was used in the calculations. The lattice constant was estimated to be 3.475 Å by using a lattice relaxation code in quantum-ESPRESSO.

A magnetic proximity effect on monolayer NbSe$_2$ was simulated by an exchange potential in a multi-orbital tight binding model describing electronic states in the pristine monolayer crystal. The tight-binding model was constructed on the basis of eleven Wannier orbitals, five *d*-orbitals in Nb atom and six *p*-orbitals in top and bottom Se atoms (see Fig. 1a) with spin degrees of freedom. Here, the hopping

integrals were computed by using Wannier90 (ref. 29), a code to provide maximally localized Wannier orbitals and hopping integrals between them from a first-principles band structure. In this model, the exchange potential was introduced as an on-site potential represented by the Zeeman coupling $H_z = -\mathbf{M} \cdot \boldsymbol{\sigma}/2$. The direction of the magnetization was fixed to the $yz$ plane with the tilting angle $\theta = \arctan\left(M_y/M_z\right)$. Here, the coupling constants $(M_y, M_z)$ are the elements of the exchange potential along the $y$ and $z$ axes, and assumed to be independent of orbital characters for simplicity. The Fermi energy $E_F$ was numerically estimated under the condition of charge neutrality between valence electrons and nuclei.

The Berry curvature was calculated by using a theoretical expression[30],

$$\Omega_n^z(\mathbf{k}) = -\sum_{n' \neq n} \frac{2\mathrm{Im}\langle\psi_{n\mathbf{k}}|v_x|\psi_{n'\mathbf{k}}\rangle\langle\psi_{n'\mathbf{k}}|v_y|\psi_{n\mathbf{k}}\rangle}{\left(\omega_{n'} - \omega_n\right)^2}, \quad (1)$$

where $|\psi_{n\mathbf{k}}\rangle$ and $E_n = \hbar\omega_n$ are the wave function and the eigen-energy, respectively, of the electronic state with the band index $n$ and the wave vector $\mathbf{k}$. The anomalous Hall conductivity ($\sigma_{xy}$) was obtained by the integration of the Berry curvatures below $E_F$ in the BZ,

$$\sigma_{xy} = -\frac{e^2}{\hbar} \int_{BZ} \frac{d^2k}{(2\pi)^2} \sum_n f_n \Omega_n^z(\mathbf{k}), \quad (2)$$

with the Fermi–Dirac distribution function $f_n$. The numerical calculation was performed at zero temperature.

## Data availability

The data within this paper are available from the corresponding author upon reasonable request.

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

## Acknowledgements

We are grateful to M. S. Bahramy and K. Ishizaka for valuable discussions, and also to Y. Wang, Y. Kashiwabara, Y. Majima, Y. Tanaka, S. Yoshida, B. K. Saika, and M. Kawasaki for experimental help. This work was supported by Grants-in-Aid for Scientific Research (Grant Nos. 19H05602, 19H02593, 19H00653, 20H01840, 20H00127, and 21K13888) and A3 Foresight Program from the Japan Society for the Promotion of Science (JSPS), and by PRESTO (Grant No. JPMJPR20AC) and CREST (Grant No. JPMJCR20T3) from Japan Science and Technology Agency.

## Author contributions

H.M. grew and characterized the samples, performed transport measurements, and analyzed the data. T.H. and M.K. contributed to theoretical interpretation. M.N. and Y.I. supervised the study. H.M., T.H., Y.I., and M.N. wrote the manuscript. All the authors discussed the results and commented on the manuscript.

## Competing interests

The authors declare no competing interests.
