## [Peer Review File · Nature Communications]

REVIEWER COMMENTS

Reviewer #1 (Remarks to the Author):

In nature communications manuscript NCOMMS-21-40268 Masuoka et al present and discuss the results of a Hall effect study of NbSe₂/V₅Se₈ heterostructures grown by molecular beam epitaxy. The authors report the observation of negative anomalous Hall effect (AHE) as the number of V₅Se₈ layers is reduced for a fixed number of NbSe₂ layers. The authors attribute the anomalous Hall effect to ferromagnetism in NbSe₂. The ferromagnetism is induced by exchange coupling of the states in the two components due to the proximity effect.

The paper is well written and contains new and interesting data. The proposed interpretation is plausible but it is not the only one. The anomalous Hall effect can appear in NbSe₂ when the carriers at the interface of the two materials undergo scattering by V₅Se₈. To unambiguously prove the proposed model one has to carry out X-ray magnetic circular dichroism measurements which yield results that are element specific.

A sign reversal of AHE is observed but its origin is not explained. Although the authors claim that sign reversal in V₅Te₈ has not been observed, they should measure their V₅Te₈ separately to confirm, as samples from different groups may have different properties. Sign reversal in V₅Te₈ or in a heterostructure will lead to different interpretations.

The theoretical calculation in the manuscript predicts AHE. Did the calculation replicate the sign reversal for different thicknesses?

Anomalous Hall effect decreases with increase field angle. Should not that be due to the trivial effect that the magnetization now lies in plane, which reduces perpendicular magnetization?

In the referee's opinion NCOMMS-21-40268 can appear in in Nature Communications provided that the authors insert a paragraph in which they state clearly that other models can explain their data as discussed above.

Reviewer #2 (Remarks to the Author):

The result of this work is interesting. The authors synthesized heterostructures of layered magnets and layered superconductors. Based on magnetic proximity effect, the heterostructure exhibits AHE effect. There are three deficiencies in my view.

1. There is no strong evidence where the AHE originates from (from NbSe₂ or VSe₂?). Both two constituent materials are metallic, and so there should be mixed contribution to the observed AHE that needs clarification.
2. There is no direct experimental evidence of ferro-valley signature, as claimed. This claim is based on calculated bandstructure. Not quite strong evidence.
3. Abstract and introduction are not clear. From abstract, it is hard to get the exact physical picture the authors want to deliver. Also, the motivation of this work described in the introduction is not strongly convincing. For example, the authors stated that the magnetic proximity effect has been studied in WSe₂/CrI₃, but not in metallic systems. Question is “why need to study metallic systems, no physics?” We understand metallic systems differ from insulating/semiconducting systems in electronic or spintronic device applications. But the authors should clearly sketch out the significance and motivation of their work.

Reviewer #3 (Remarks to the Author):

In this paper, the authors tuned an originally-superconducting two-dimensional NbSe₂ into a ferromagnetic/ferrovalley state with spontaneous spin-valley polarization by interfacing with a two-dimensional ferromagnet VSe₂. They further investigated the anomalous Hall effect (AHE) of the NbSe₂/VSe₂ magnetic vdW heterostructures, and demonstrated that the sign of the AHE was reversed with the number of the VSe₂ layer was thinned down to the monolayer limit. Such topic is very interesting and important for the fundamental research in the condensed matter physics and materials science. I would recommend its publication on NC after addressing the following important issues.

1. One important issue: the authors stated that 2D NbSe₂ is a nonmagnetic material without interfacing with ferromagnet VSe₂. So it is not an intrinsic magnetic material? However, to my knowledge, there are many previous works on 2D NbSe₂ [i.e., Nano Res. 2021, 14, 834 ; Angew. Chem. Int. Ed. 2017, 56, 10214 ; ACS Nano 2012, 6, 11, 9727] reported that it is an intrinsic magnetic material. The authors should address this discrepancy. Is it from CDW?
2. The magnitude of spin splitting energy in Fig. 3e is not so straightforward to express the different absolute values between K and K' are different.
3. How did the authors check the external magnetic field of 40 meV? Any reasons? Will the variation of external magnetic field affect the results?

Responses to Reviewer #1

Comment #1-1:

In nature communications manuscript NCOMMS-21-40268 Masuoka et al present and discuss the results of a Hall effect study of NbSe₂/V₅Se₈ heterostructures grown by molecular beam epitaxy. The authors report the observation of negative anomalous Hall effect (AHE) as the number of V₅Se₈ layers is reduced for a fixed number of NbSe₂ layers. The authors attribute the anomalous Hall effect to ferromagnetism in NbSe₂. The ferromagnetism is induced by exchange coupling of the states in the two components due to the proximity effect.

Response #1-1:

We would appreciate the reviewer for spending his/her invaluable time to review our manuscript. There seem some misunderstandings of our observations and interpretation. We note that negative anomalous Hall effect (AHE) was observed for the heterostructure samples with thick-enough V₅Se₈ (as well as for the V₅Se₈ individual films), while positive AHE was observed for the samples with thin-enough V₅Se₈. We attribute this positive AHE to ferromagnetism in NbSe₂ induced by proximity coupling with ferromagnetic V₅Se₈ at the van der Waals interface.

Comment #1-2:

The paper is well written and contains new and interesting data. The proposed interpretation is plausible but it is not the only one. The anomalous Hall effect can appear in NbSe₂ when the carriers at the interface of the two materials undergo scattering by V₅Se₈. To unambiguously prove the proposed model one has to carry out X-ray magnetic circular dichroism measurements which yield results that are element specific.

Response #1-2:

We are happy to see the reviewer's comment that "The paper is well written and contains new and interesting data. The proposed interpretation is plausible". We agree to the comment that x-ray magnetic circular dichroism (XMCD) measurements should provide strong proof of our model. However, we argue that XMCD measurements on Nb are extremely challenging because of the limited photon energy available in the synchrotron facilities, and in fact no successful XMCD results on Nb have been reported so far. We also note that ferromagnetism should be induced only in the single layer NbSe₂ in contact with V₅Se₈, and the induced magnetic moment should be very small, which also makes it difficult to obtain a sizable XMCD signal from this heterostructure system.

The reviewer has concerned about our interpretation of the observed experimental results, and raised other possible scenarios. However, we insist that our interpretation is the only one that could consistently explain our observations. Since there seem some misunderstandings of our claims judging from **Comment #1-1**, we here summarize our observations and interpretation.

- I. The NbSe₂/V₅Se₈ heterostructure samples show negative AHE when V₅Se₈ is thick-enough. This is simply because the electrical conductions of those samples are dominated by the thick

V₅Se₈ metallic layer, which is ferromagnetic and shows negative AHE as we reported in the previous study [M. Nakano, *et al.*, *Nano Lett.* **19**, 8806 (2019)]. The behavior of the V₅Se₈ individual films will be discussed later in **Response #1-3** more in detail.

- II. The heterostructure samples start to exhibit **positive AHE when V₅Se₈ becomes thin-enough.** Considering that V₅Se₈ becomes insulating at this thin-enough limit (this is also discussed in the previous study cited above as well as in the main text and Supplementary Information of the present study), the electrical conduction of those samples should be governed by the 4 L-thick NbSe₂ layer, and the observed **positive AHE should be attributed to ferromagnetism in NbSe₂** induced by proximity coupling with neighboring V₅Se₈.
- III. Those heterostructure samples characterized with positive AHE demonstrate **enhancement of AHE against the in-plane magnetic fields**, which has never been observed in other systems.
- IV. Theoretical calculations based on the band structure of ferromagnetically-proximitized NbSe₂ successfully **reproduce both positive AHE and enhancement of AHE against the in-plane magnetic fields**, unambiguously proving that NbSe₂ forms a ferromagnetic state at the NbSe₂/V₅Se₈ van der Waals interface.

The sign reversal of AHE could be potentially explained by a scattering process at the interface with V₅Se₈ as the reviewer pointed out and/or by the conventional Skew scattering or side-jump scattering mechanisms. However, we emphasize that the AHE signal generated by such a scattering event should show a simple behavior against the in-plane magnetic fields, and the observed **enhancement of AHE against the in-plane magnetic fields could not be explained by those scattering mechanisms.**

The contents I-IV shown above are already written in the main text and in Supplementary Information. However, it seems that the logic flow was not clear enough. In particular, the motivation of the angle-dependence measurements was not well written. We largely revised the manuscript so that all the readers could easily follow our arguments written above.

Comment #1-3:

A sign reversal of AHE is observed but its origin is not explained. Although the authors claim that sign reversal in V₅Te₈ has not been observed, they should measure their V₅Te₈ separately to confirm, as samples from different groups may have different properties. Sign reversal in V₅Te₈ or in a heterostructure will lead to different interpretations.

Response #1-3:

Thank you for the comments. As we explain above in **Response #1-2**, the sign reversal of AHE from negative to positive should be originating from the fact that the dominant layer providing larger contribution to the electrical conduction is varied from V₅Se₈ to NbSe₂ as the number of the V₅Se₈ layer is reduced from the thick-enough regime to the thin-enough limit. This is already written in the main text of the present manuscript, but we revised the manuscript to make it clearer. As for the behavior of the V₅Se₈ individual films, we confirmed in our previous study [M. Nakano, *et al.*, *Nano Lett.* **19**, 8806 (2019)] that **our V₅Se₈ (not V₅Te₈) thin film samples do not show the sign reversal of AHE down to the 2D limit.** Figure R1-1 shows the same data as those shown in the previous paper, corresponding to (a) the magnetic-field-dependence data at $T = 2$ K and (b) the temperature-

dependence data at $\mu_0H = +3$ T for three representative samples with different layer numbers. All the samples show negative AHE at all temperatures, and the sign reversal has never been observed. Those results should exclude a possibility that the sign reversal of AHE observed in our NbSe₂/V₅Se₈ heterostructure samples is originating from that in the V₅Se₈ individual films.

Figure R1-1. a, The ρ_{AH} versus μ_0H curves of the 30 L- (red), 8 L- (green), and 3 L- (blue) thick V₅Se₈ individual films at $T = 2$ K. The data of the 30 L- and 3 L-thick films are vertically shifted by 0.1 and -0.1 $\mu\Omega\text{cm}$, respectively. **b**, The temperature dependences of ρ_{AH} of the same samples as shown in (a) at the saturated field regime, $\mu_0H = +3$ T.

Comment #1-4:

The theoretical calculation in the manuscript predicts AHE. Did the calculation replicate the sign reversal for different thicknesses?

Response #1-4:

As we explain above, the sign of AHE is governed by the dominant layer providing larger contribution to the electrical conduction; negative AHE should appear when the electrical conduction is dominated by ferromagnetic V₅Se₈ (*i.e.*, when V₅Se₈ is thick-enough), while positive AHE should appear when the electrical conduction is dominated by ferromagnetic NbSe₂ (*i.e.*, when V₅Se₈ is thin-enough). The sign reversal that we observed in our experiments should be associated with such a parallel conduction effect rather than an interface effect. In this study, we calculated the contribution from ferromagnetic NbSe₂ and did not consider the contribution from ferromagnetic V₅Se₈. This is because we focused on the AHE of the samples with thin-enough V₅Se₈ in our experiments, where the contribution from V₅Se₈ should be negligible. However, if we extend our theoretical model and consider the contribution from ferromagnetic V₅Se₈, we should be able to reproduce the sign reversal as the number of the V₅Se₈ layer is reduced. We would like to treat this as an interesting future work.

Comment #1-5:

Anomalous Hall effect decreases with increase field angle. Should not that be due to the trivial effect that the magnetization now lies in plane, which reduces perpendicular magnetization? In the referee's opinion NCOMMS-21-40268 can appear in in Nature Communications provided that the authors

insert a paragraph in which they state clearly that other models can explain their data as discussed above.

Response #1-5:

As the reviewer pointed out, the AHE signal should decrease when the field angle is changed from the out-of-plane direction to the in-plane direction. This is because the AHE signal is usually proportional to the out-of-plane component of the total magnetization, and in such a case, the signal should decrease by following $\cos(\theta)$ with the field angle θ . In our case, however, **the AHE signal does not follow $\cos(\theta)$** . This is the most unique aspect of the current system. This behavior could not be explained by simple tilting of the magnetization, but successfully explained by the emergence of the Berry curvature with the in-plane magnetic fields, which is associated with a unique feature of a ferromagnetic state in 2D NbSe₂ with Zeeman-type spin-orbit interaction. In our opinion, **other scenarios that the reviewer pointed out could not explain our data in particular for the results of the angle-dependence measurements**. We largely revised the manuscript so that the readers could understand this very important and significant point of the current study more easily.

Responses to Reviewer #2

General Comments:

The result of this work is interesting. The authors synthesized heterostructures of layered magnets and layered superconductors. Based on magnetic proximity effect, the heterostructure exhibits AHE effect. There are three deficiencies in my view.

Response:

We would appreciate the reviewer for spending his/her invaluable time to review our manuscript. We are happy to see that the reviewer finds our study interesting. Here below we would like to address three points one-by-one as follows.

Comment #2-1:

1. There is no strong evidence where the AHE originates from (from NbSe₂ or V₅Se₈?). Both two constituent materials are metallic, and so there should be mixed contribution to the observed AHE that needs clarification.

Response #2-1:

Thank you for an important comment. We agree that there is parallel conduction in the V₅Se₈ and NbSe₂ layers since both are metallic. Now that the number of the NbSe₂ layer is fixed to 4 L, we could naively consider that the dominant layer providing larger contribution to the electrical conduction is varied from V₅Se₈ to NbSe₂ as the number of the V₅Se₈ layer is reduced from the thick-enough regime above 4 L to the thin-enough regime below 4 L. Given these situations, it is natural to consider that *negative* AHE observed with **thick-enough V₅Se₈** should be attributed to **originally-ferromagnetic V₅Se₈**, while *positive* AHE observed with **thin-enough V₅Se₈** should be attributed to **ferromagnetically-proximitized NbSe₂**. Moreover, we confirmed in our previous study [M. Nakano, *et al.*, *Nano Lett.* **19**, 8806 (2019)] that the 3 L-thick V₅Se₈ individual film exhibits insulating behavior at low temperature, supporting our claim that the electrical conduction of the samples with thin-enough V₅Se₈ should be predominantly governed by NbSe₂ and that positive AHE observed in those samples should be originating from ferromagnetic NbSe₂. These interpretations are already written in the main text and in Supplementary Information, but we revised the manuscript so that all the readers could clearly and more easily follow our logic written above.

Comment #2-2:

2. There is no direct experimental evidence of ferro-valley signature, as claimed. This claim is based on calculated bandstructure. Not quite strong evidence.

Response #2-2:

Thank you for the comment. We agree that we do not provide direct evidence of the formation of a “ferrovalley” state, but this is a natural consequence of the spin-valley locking effect in *H*-type TMDCs associated with Zeeman-type SOI. This spin-valley locking effect should be relevant here if ferromagnetism in NbSe₂ is originating from the modulation of the band structure by the proximity

effect, and a good agreement between experiments and band structure calculations strongly supports this scenario. In particular, the unique angle-dependence of the AHE could not be explained by the extrinsic scattering mechanisms, suggesting that ferromagnetism in NbSe₂ should be originating from its band structure, and therefore, accompanying a ferrovalley state. We revised the corresponding part in the main text by putting the characteristic word “the spin-valley locking effect” so that the readers could more naturally and intuitively understand the inseparable relationship between a ferromagnetic state and a ferrovalley state in *H*-type TMDCs.

Comment #2-3:

3. Abstract and introduction are not clear. From abstract, it is hard to get the exact physical picture the authors want to deliver. Also, the motivation of this work described in the introduction is not strongly convincing. For example, the authors stated that the magnetic proximity effect has been studied in WSe₂/CrI₃, but not in metallic systems. Question is “why need to study metallic systems, no physics?” We understand metallic systems differ from insulating/semiconducting systems in electronic or spintronic device applications. But the authors should clearly sketched out the significance and motivation of their work.

Response #2-3:

Thank you for the very important comments. We now understand that we could not properly introduce the motivation and significance of the present study, in particular the reason why we need to study a metallic system, in the previous manuscript. There are in fact many essential differences between metallic and semiconducting systems in the context of a magnetic proximity effect. Here below we describe examples of such differences both from the device application and the fundamental physics viewpoints, and clarify the significance of the current study.

The first fundamental difference which is important from the device application viewpoint is that **a metallic system should host a net magnetization at a ground state, whereas a semiconducting system should not.** In our NbSe₂/V₅Se₈ heterostructures, for example, proximitized NbSe₂ should have a net magnetization at a ferromagnetic ground state. This is simply because the numbers of the up-spin electrons and the down-spin electrons below E_F should become different when proximitized by V₅Se₈. In a semiconducting system such as the WSe₂/CrI₃ heterostructures, on the other hand, proximitized WSe₂ should have a modulation of its electronic band structure under the presence of the exchange field from CrI₃, but it should not have a net magnetization at a ground state. This is because WSe₂ is a semiconductor characterized with the fully-occupied bands, where the numbers of the up-spin electrons and the down-spin electrons below E_F should be always equal even when proximitized by CrI₃. This is a striking difference between the present study on a metallic system and the previous study on a semiconducting system, and realization of a ferromagnetic ground state with spontaneous spin polarization in NbSe₂ is one of the most significant parts of the present study.

The second difference which is important from the fundamental physics viewpoint is that **a metallic system should enable us to examine the low-energy electronic properties of the system by transport measurements, whereas a semiconducting system should not.** In particular, the AHE could be a very powerful probe, through which we should be able to obtain fundamental information

on the electronic structure of the system near E_F . In the present study, we could in fact verify an interplay between magnetism and Zeeman-type SOI in a non-centrosymmetric 2D quantum material for the first time through the AHE measurements. As we wrote in the manuscript, we observed intriguing enhancement of AHE against the in-plane magnetic fields, which could be attributed to the emergence of the Berry curvature along the spin-degenerate nodal lines in 2D NbSe₂ by the in-plane magnetization. Such a unique AHE signal that is not proportional to magnetization has never been observed in 2D quantum material systems so far. Moreover, this behavior is clearly distinct from those discussed in topological systems in terms of symmetry, providing insights into a link between those two big research fields. We believe that this is another significant part of the present study.

We largely revised the manuscript including the abstract and introduction parts to properly introduce the motivation and significance of the present study described above. We believe that now the quality of the paper has been largely improved. Thank you very much for the constructive comments.

Responses to Reviewer #3

General Comments:

In this paper, the authors tuned an originally-superconducting two-dimensional NbSe₂ into a ferromagnetic/ferrovalley state with spontaneous spin-valley polarization by interfacing with a two-dimensional ferromagnet V₅Se₈. They further investigated the anomalous Hall effect (AHE) of the NbSe₂/V₅Se₈ magnetic vdW heterostructures, and demonstrated that the sign of the AHE was reversed with the number of the V₅Se₈ layer was thinned down to the monolayer limit. Such topic is very interesting and important for the fundamental research in the condensed matter physics and materials science. I would recommend its publication on NC after addressing the following important issues.

Response:

We would appreciate the reviewer for spending his/her invaluable time to review our manuscript. We are very happy to see that the reviewer finds our work of the utmost quality with the comment that “Such topic is very interesting and important for the fundamental research in the condensed matter physics and materials science” and recommends its publication in Nature Communications. Here below we would give our responses to the comments.

Comment #3-1:

1. One important issue: the authors stated that 2D NbSe₂ is a nonmagnetic material without interfacing with ferromagnet V₅Se₈. So it is not an intrinsic magnetic material? However, to my knowledge, there are many previous works on 2D NbSe₂ [i.e., *Nano Res.* 2021, 14, 834; *Angew. Chem. Int. Ed.* 2017, 56, 10214; *ACS Nano* 2012, 6, 11, 9727] reported that it is an intrinsic magnetic material. The authors should address this discrepancy. Is it from CDW?

Response #3-1:

Thank you for the comments. We argue that 2D NbSe₂ is not an intrinsic magnetic material. As the reviewer pointed out, there are many studies claiming 2D NbSe₂ to be an intrinsic magnetic material, but these are all **theoretical** works. On the other hand, **experimental** studies from several independent groups have already proven that 2D NbSe₂ is not a magnetic material but a superconductor down to the monolayer limit [for examples, see X. Xi, *et al.*, *Nat. Phys.* **12**, 139 (2016), M. M. Ugeda, *et al.*, *Nat. Phys.* **12**, 92 (2016), Y. Xing, *et al.*, *Nano Lett.* **17**, 6802 (2017)]. Importantly, we confirmed that our 2D NbSe₂ individual films grown by our MBE system are also superconducting down to the monolayer limit [H. Matsuoka, *et al.*, *Phys. Rev. Research* **2**, 012064(R) (2020)]. Those experimental studies clearly exclude a possibility that 2D NbSe₂ is an intrinsic magnetic material. Here below we provide short summaries and comments on possible reasons for the discrepancy between theory and experiment for the papers cited by the reviewer above.

1. *Nano Res.* 2021, 14, 834: This theoretical paper predicts monolayer NbX₂ (X = S, Se) to be ferromagnetic/ferrovalley due to their intrinsic magnetic exchange interaction and inversion asymmetry. The claim is similar to that of the reference 24 that we cited in the present study.

Those studies however fail to reproduce the experimental results, presumably due to overestimation of the exchange interactions.

2. *Angew. Chem. Int. Ed.* 2017, 56, 10214: This theoretical paper predicts monolayer 1S-NbX₂ (X = S, Se, Te) to be a diamagnetic direct-gap semiconductor. We note that diamagnetism should be a character of a “non-magnetic” material, where the long-range magnetic order is absent. Moreover, a polytype that we are interested in is not S-type (Haeckelite type) but H-type (trigonal prismatic), and therefore, this paper should be irrelevant to our research.
3. *ACS Nano* 2012, 6, 11, 9727: This theoretical paper predicts the biaxial tensile strained NbS₂ and NbSe₂ to be a ferromagnet with very high Curie temperature exceeding room temperature. However, a magnitude of strain required to realize such a ferromagnetic state is as high as a few percent level, which is unrealistic in the present case since the interface of the NbSe₂/V₅Se₈ heterostructure is made of a weak van der Waals bonding. We confirmed by experiments and described in the previous study [H. Matsuoka, *et al.*, *Nano Lett.* **21**, 1807 (2021)] that the epitaxial strain in this system is in fact negligible.

Comment #3-2:

2.The magnitude of spin splitting energy in Fig. 3e is not so straightforward to express the different absolute values between K and K' are different.

Response #3-2:

Thank you for the comment. We agree that the sentence “the magnitude of $\Delta_{\text{up-down}}$ at the K and K' valleys becomes different (see Fig. 3e)” is misleading. This could not be clearly seen in Fig. 3e, but in Fig. 3d (now changed to Fig. 4d). We modified the corresponding sentence to be “the magnitude of $\Delta_{\text{up-down}}$ at the K and K' valleys becomes different (see Fig. 4d)”. Thank you very much.

Comment #3-3:

3.How did the authors check the external magnetic field of 40 meV? Any reasons? Will the variation of external magnetic field affect the results?

Response #3-3:

Thank you for an important comment. We discuss the calculation results with the fixed exchange field (not “external magnetic field”) $|M| = 40$ meV in the main text, but we confirm that **we could obtain essentially the same results [positive AHE and enhancement of AHE against the in-plane magnetic fields] for a rather broad range of the exchange fields from a few millielectronvolt to a hundred millielectronvolt**. We discuss this important point in Supplementary Information in detail. Figure R3-1 shows the same data as those shown in Supplementary Information section F as Fig. S6, corresponding to (a) the angle-dependence of the AH conductivity (σ_{xy}) with different $|M|$ and (b) the magnitude of a deviation of σ_{xy} from $\cos(\theta)$ at $\theta = 20^\circ$ as a function of $|M|$. The largest deviation from $\cos(\theta)$ behavior is achieved when $|M| = 40$ meV, but a deviation could be seen for a broad range of $|M|$. As we discuss in the main text, the exchange field as large as a few tens of millielectronvolt scale is consistent to those estimated for other van der Waals systems including WSe₂- and WS₂-based magnetic heterostructures.

Figure R3-1. a, The angle dependence of the σ_{xy} with different $|M|$ at $E = E_F$ calculated from the band structure of monolayer NbSe₂. The configuration of θ is shown in the inset. **b**, The magnitude of a deviation of the σ_{xy} from $\cos(\theta)$ at $\theta = 20^\circ$ as a function of $|M|$.

REVIEWER COMMENTS

Reviewer #1 (Remarks to the Author):

This referee is satisfied that the authors have addressed in a satisfactory manner all the issues raised in his report. Therefore the referee is of the opinion that NCOMMS-21-40268A can be published

Reviewer #2 (Remarks to the Author):

I read all referee's comments and questions, and jump out to read the main results and conclusions again. I am hesitating to accept the conclusion that the authors experimentally observed the magnetized NbSe₂ (due to proximity coupling to V₅Se₈). Regarding my first comment, the authors' reply is not strong and convincing. Figure S2 showed the clear electrical resistivity difference between the two materials of similar thickness. So, the authors' simple explanation in response to my question #1 regarding how to distinguish the electrical conduction of the two materials appears still obscure. Transport is a complex phenomenon. As pointed out by the referee #1, the hysteresis loop could be due to the spin dependent scattering from V₅Se₈. The change of AHE sign could be related to the detailed band properties when sample thins. It is coarse to directly explain the positive and negative AHE to the respective contribution of the two different materials.

Overall, I did not find strong evidence of magnetism "in NbSe₂". Experimental results of this work appear not rich/strong enough to clarify the above concern and the similar concern by referee #1. I encourage the authors to extend more experimental investigation to nail down these puzzles.

Reviewer #3 (Remarks to the Author):

Concerning Nb, it has five valence electrons. In NbX₂, four electrons are used for the Nb-X bond. So the left one electron would result in a magnetic moment on Nb.

Since the authors claim that it is nonmagnetic. Then the authors should explain where the left electron go.

Responses to Reviewer #1

Comment:

This referee is satisfied that the authors have addressed in a satisfactory manner all the issues raised in his report. Therefore the referee is of the opinion that NCOMMS-21-40268A can be published.

Response:

We would appreciate your effort for reviewing our manuscript again. We are very happy to see that the reviewer finds our responses satisfactory and recommends publication in *Nature Communications*. We believe that our manuscript has been largely improved thanks to your very constructive comments. Thank you very much.

Responses to Reviewer #2

Comment:

I read all referee's comments and questions, and jump out to read the main results and conclusions again. I am hesitating to accept the conclusion that the authors experimentally observed the magnetized NbSe₂ (due to proximity coupling to V₅Se₈). Regarding my first comment, the authors' reply is not strong and convincing. Figure S2 showed the clear electrical resistivity difference between the two materials of similar thickness. So, the authors' simple explanation in response to my question #1 regarding how to distinguish the electrical conduction of the two materials appears still obscure. Transport is a complex phenomenon. As pointed out by the referee #1, the hysteresis loop could be due to the spin dependent scattering from V₅Se₈. The change of AHE sign could be related to the detailed band properties when sample thins. It is coarse to directly explain the positive and negative AHE to the respective contribution of the two different materials.

Overall, I did not find strong evidence of magnetism "in NbSe₂". Experimental results of this work appear not rich/strong enough to clarify the above concern and the similar concern by referee #1. I encourage the authors to extend more experimental investigation to nail down these puzzles.

Response:

We would appreciate your effort for reviewing our manuscript again. First of all, as we mentioned in the previous Response Letter, we argue that the strong evidence that supports our main claim in this paper (formation of a ferromagnetic ground state in NbSe₂) is **NOT** the sign reversal of AHE **BUT** the unique angle-dependence of AHE characterized with the enhancement of AHE against the in-plane magnetic fields, which could be explained only if we accept the situation that NbSe₂ forms a ferromagnetic state at the interface. As we wrote in Responses #2-2, #1-2 and #1-5 in the previous Response Letter, "a spin-dependent scattering" at the interface could potentially explain the sign reversal of AHE, but it could not explain the enhancement of AHE against the in-plane fields.

In order to emphasize this important point, we newly added the sentence "*We emphasize that the unique angle dependence of the AHE characterized with the enhancement of the AHE signal with the in-plane fields could be well reproduced by theoretical calculations based on the band structure of ferromagnetic NbSe₂, providing firm evidence that NbSe₂ forms a ferromagnetic ground state at the interface with V₅Se₈.*" in the Discussion section of the revised manuscript.

Regarding the first comment, we believe that the electrical conduction of the particular samples that we are interested in (*i.e.*, the heterostructure samples with *thin-enough* V₅Se₈) is dominated by NbSe₂ rather than V₅Se₈ simply because V₅Se₈ should be insulating in this regime ($N < 2.0$ L) while 4 L-thick NbSe₂ should be still metallic. The reviewer wrote "*Figure S2 showed the clear electrical resistivity difference between the two materials of similar thickness. So, the authors' simple explanation in response to my question #1 regarding how to distinguish the electrical conduction of the two materials appears still obscure.*", but Fig. S2 just evidences that V₅Se₈ becomes insulating while NbSe₂ remains metallic in the thin-enough regime, supporting our claim above. We note that

Reviewer #1 had a similar concern in the previous round, but he/she is satisfied with our replies and now supports publication of this study in *Nature Communications*.

We believe that the above simple explanation based on the longitudinal resistivity should be enough to support our claim that the electrical conduction of the heterostructure samples with *thin*-enough V_5Se_8 is dominated by $NbSe_2$, but here we provide a new set of data to further support this claim. Figure R2-1 shows the Hall coefficient R_H of the heterostructure samples at $T = 2$ K as a function of the number of the V_5Se_8 layer. We find that the heterostructure samples with *thin*-enough V_5Se_8 show clear positive R_H , which is close to the R_H value of a typical $NbSe_2$ film grown by our MBE system. Given that the R_H value of the V_5Se_8 individual films stays negative down to the 2D limit as we reported in the previous study [M. Nakano, *et al.*, *Nano Lett.* **19**, 8806 (2019)], the obtained results support our claim that the electrical conduction in this regime is dominated by $NbSe_2$.

Figure R2-1. The R_H value of the heterostructure samples at $T = 2$ K as a function of the number of the V_5Se_8 layer. Those of the V_5Se_8 individual films reported in our previous study [M. Nakano, *et al.*, *Nano Lett.* **19**, 8806 (2019)] and that of a typical $NbSe_2$ film are also shown for reference.

As for the comment “*The change of AHE sign could be related to the detailed band properties when sample thins.*”, we exclude this possibility because the V_5Se_8 individual films do not show the sign reversal of AHE down to the 2D limit as we confirmed in the previous study [M. Nakano, *et al.*, *Nano Lett.* **19**, 8806 (2019)]. Figure R2-2 shows the same data as those shown in the previous paper, corresponding to (a) the magnetic-field-dependence data at $T = 2$ K and (b) the temperature-dependence data at $\mu_0H = +3$ T for three representative V_5Se_8 individual films with different layer numbers. All the samples show negative AHE at all temperatures, and the sign reversal has never been observed. Those results should exclude a possibility that the sign reversal of AHE observed in our $NbSe_2/V_5Se_8$ heterostructure samples is originating from that in the V_5Se_8 individual films. The same explanation was given in Response #1-3 in the previous Response Letter.

Figure R2-2. a, The ρ_{AH} versus $\mu_0 H$ curves of the 30 L- (red), 8 L- (green), and 3 L- (blue) thick $V_5\text{Se}_8$ individual films at $T = 2$ K. The data of the 30 L- and 3 L-thick films are vertically shifted by 0.1 and -0.1 $\mu\Omega\text{cm}$, respectively. **b**, The temperature dependences of ρ_{AH} of the same samples as shown in (a) at the saturated field regime, $\mu_0 H = +3$ T.

Responses to Reviewer #3

Comment:

Concerning Nb, it has five valence electrons. In NbX₂, four electrons are used for the Nb-X bond. So the left one electron would result in a magnetic moment on Nb. Since the authors claim that it is nonmagnetic. Then the authors should explain where the left electron goes.

Response:

Thank you for the comment. As you wrote, there is one electron per Nb left without formation of the Nb-X bond when considering the charge neutrality of NbX₂ based on the ionic model. However, this “excess” electron is delocalized in the entire crystal as a free electron to form an electronic band instead of being localized on Nb to form a local magnetic moment. Whether the “excess” electron is delocalized or localized is in fact a very subtle issue, which is determined by a competition between the kinetic energy gain and the Coulomb energy cost through a hopping process between neighboring atoms (*i.e.*, a competition between the band width W and the on-site Coulomb repulsion energy U). In solid state physics, it is generally considered that $3d$ electrons tend to be localized due to their smaller spatial distribution (smaller W), often leading to the formation of a local magnetic moment on each atom. On the other hand, $4d$ electrons including the “excess” electron in NbSe₂ tend to be delocalized due to their larger spatial distribution (larger W), usually leading to the formation of a dispersive electronic band. In such a case, there is no net magnetic moment formed on each atom, and the system does not host intrinsic magnetism.

REVIEWERS' COMMENTS

Reviewer #2 (Remarks to the Author):

I appreciate the authors' detailed response. The angle dependent data can deliver stronger evidence for proximity induced magnetism. As I commented in the first round of review, claiming anything related valley in this work is a little too much, since there is no experimental proof of valley related properties. The authors demonstrated spin-related evidence, and use band structure calculation to link to "valley". This, in some occasions, is true, for hexagonal lattice with strong SOC. However, as an experiment-focused work (supported by calculations), claiming spin polarization is proper, but claiming valley polarization does not have any experimental evidence support. I encourage the authors to revise to make the manuscript more fitting to its actual content.

Reviewer #3 (Remarks to the Author):

All my concerns are well addressed. The corresponding discussions on the localization or delocalization of d electrons must be included in the main manuscript.

Responses to Reviewer #2

Comment:

I appreciate the authors' detailed response. The angle dependent data can deliver stronger evidence for proximity induced magnetism. As I commented in the first round of review, claiming anything related valley in this work is a little too much, since there is no experimental proof of valley related properties. The authors demonstrated spin-related evidence, and use band structure calculation to link to "valley". This, in some occasions, is true, for hexagonal lattice with strong SOC. However, as an experiment-focused work (supported by calculations), claiming spin polarization is proper, but claiming valley polarization does not have any experimental evidence support. I encourage the authors to revise to make the manuscript more fitting to its actual content.

Response:

We would deeply appreciate your continuing effort for reviewing our manuscript. We are very happy to see that the reviewer is now convinced by our explanation that the results of the angle dependence of AHE strongly support our main claim, the formation of a ferromagnetic ground state in NbSe₂. Regarding the formation of a ferrovalley state, we still believe that this should be the case given the strong spin-valley locking effect in NbSe₂ as we wrote in the first Response Letter. On the other hand, we agree that there is no direct experimental evidence of the formation of a ferrovalley state provided in this study. We revised the manuscript to tone down the valley polarization related claims. Thank you again for giving us many constructive comments through the whole review process.

Responses to Reviewer #3

Comment:

All my concerns are well addressed. The corresponding discussions on the localization or delocalization of d electrons must be included in the main manuscript.

Response:

We are happy to see that the reviewer's concerns are well addressed. According to your comment, we revised the sentence explaining the basic properties of NbSe₂ in the second paragraph in page 4 from “NbSe₂ is one of representative metallic *H*-type TMDCs showing the charge-density wave (CDW) and the superconducting (SC) transition at low temperature¹⁴.” to “NbSe₂ is one of representative metallic *H*-type TMDCs showing the charge-density wave (CDW) and the superconducting (SC) transition at low temperature¹⁴, while it is magnetically inactive due to highly delocalized nature of *4d* electrons in NbSe₂.”. Thank you for the comment.